# Drug Repurposing in the Chemotherapy of Infectious Diseases

**DOI:** 10.3390/molecules29030635

**Published:** 2024-01-29

**Authors:** Amal Hamid, Pascal Mäser, Abdelhalim Babiker Mahmoud

**Affiliations:** 1Faculty of Pharmacy, University of Khartoum, Khartoum 11111, Sudan; amlhm_92@hotmail.com; 2Department of Medical Parasitology and Infection Biology, Swiss Tropical and Public Health Institute, Allschwil, 4123 Basel, Switzerland; 3Faculty of Science, University of Basel, 4001 Basel, Switzerland; 4Department of Microbial Natural Products, Helmholtz Institute for Pharmaceutical Research Saarland, 66123 Saarbruecken, Germany; 5Department of Microbial Drugs, Helmholtz Centre for Infection Research (HZI), 38124 Braunschweig, Germany

**Keywords:** drug research and development, drug repurposing, drug repositioning, drug target, infectious disease, neglected tropical disease, chemotherapy

## Abstract

Repurposing is a universal mechanism for innovation, from the evolution of feathers to the invention of Velcro tape. Repurposing is particularly attractive for drug development, given that it costs more than a billion dollars and takes longer than ten years to make a new drug from scratch. The COVID-19 pandemic has triggered a large number of drug repurposing activities. At the same time, it has highlighted potential pitfalls, in particular when concessions are made to the target product profile. Here, we discuss the pros and cons of drug repurposing for infectious diseases and analyze different ways of repurposing. We distinguish between opportunistic and rational approaches, i.e., just saving time and money by screening compounds that are already approved versus repurposing based on a particular target that is common to different pathogens. The latter can be further distinguished into divergent and convergent: points of attack that are divergent share common ancestry (e.g., prokaryotic targets in the apicoplast of malaria parasites), whereas those that are convergent arise from a shared lifestyle (e.g., the susceptibility of bacteria, parasites, and tumor cells to antifolates due to their high rate of DNA synthesis). We illustrate how such different scenarios can be capitalized on by using examples of drugs that have been repurposed to, from, or within the field of anti-infective chemotherapy.

## 1. Repurposing in Nature

Repurposing is a ubiquitous mechanism in Nature. It plays a significant role in the process of evolution that involves the adaptation of existing biological structures, functions, or traits to serve new or additional purposes [1]. Illustrative examples are feathers, which originally evolved for insulation and were later repurposed for flight [2]; the repurposing of the reptilian jaw to the mammalian auditory ossicles [3]; and the repurposing of the primary cilium of opisthokonts as a sensory organelle [4]. Repurposing also involves modifying the structure and activity of existing enzymes to perform novel or enhanced functions, harnessing the versatility and catalytic power of enzymes for applications beyond their original roles. This is exemplified by transceptors, proteins that possess both transport and sensing functions [5], and also by the moonlighting functions of many enzymes [6]. In *Plasmodium vivax*, the glycolytic enzyme aldolase also induces the motility of the parasite and plays a crucial role in host cell invasion [7]. Moreover, gene duplication events followed by functional divergence can lead to the emergence of new or expanded functions. For instance, the evolution of hemoglobin paralogues from a single ancestral globin gene allowed for oxygen transport in both respiratory and muscular tissues, and oxygen transport from mother to fetus [8].

Chemical repurposing also occurs in Nature, whereby naturally occurring compounds, originally evolved for one function, are subsequently utilized for different functions in biological systems. This process allows for the adaptation and exploitation of pre-existing chemical resources to serve novel biological roles. Secondary metabolites are a typical example. They are produced by organisms to serve as chemical defenses against predators, competitors, or pathogens, and have frequently been repurposed for other functions such as signaling, communication, or symbiotic interactions, allowing for versatility and ecological adaptability [9]. Chemical repurposing also includes the repurposing of proteinogenic amino acids as neurotransmitters (e.g., glycine, glutamate, and aspartate), as well as the repurposing of DNA for neutrophil extracellular traps (NETs), chromatin structures that can trap and kill bacteria, fungi, and protozoa as a first-line defense against pathogens [10]. Thus, repurposing is a natural strategy for innovation.

## 2. Repurposing in Drug Development

Repurposing is also imperative in the process of invention of human technology, in particular with respect to drug development. Since the development of a new drug is lengthy and expensive, repurposing (i.e., finding new uses for existing drugs) is a strategy that offers an attractive shortcut between the bench and the clinic, particularly where the resources for R&D are limited [11,12,13,14]. The very origin of chemotherapy lies in repurposing: quinine’s repurposing as an antimalarial drug marked one of the earliest instances of drug repurposing. The alkaloid extracted from the bark of the Cinchona tree had been used for centuries by indigenous populations in South America to treat fevers and shivering [15]. In the 17th century, quinine was introduced to Europe and became the primary treatment for malaria [16]. The unique case of the antimalarial quinine represented a milestone in the history of drug discovery where the drug (quinine) was discovered before the differentiation and elaboration of the disease itself [17].

In the late 19th century, Paul Ehrlich discovered that certain synthetic dyes used in the emerging textile industry preferentially stained pathogens over their host cells. If selective staining is possible, he reasoned, so must selective chemotherapy be. Building upon this work, Ehrlich and his colleague Sahachiro Hata developed salvarsan, an arsenic-based compound, as a treatment for syphilis. Salvarsan represented one of the earliest examples of a chemotherapeutic agent used to target a specific pathogen [18]. Methylene blue, a synthetic dye commonly used in various applications, was repurposed as an antimalarial drug and used for the treatment of malaria [19]. A few years later, and based on Ehrlich’s work on trypanocidal dyes, suramin was introduced for the treatment of African trypanosomiasis [20]. Suramin was later repurposed for onchocerciasis [21]. The sulfonamides, p-aminobenzoic acid analogs that block folate synthesis by inhibiting dihydropteroate synthase, originated from the red dye prontosil [22]. These historical examples illustrate how the repurposing of synthetic dyes led to the discovery and development of anti-infective drugs that formed the foundation of modern chemotherapy. They highlight the early understanding that certain chemicals, originally designed for non-medical purposes, selectively bind to pathogens, opening a therapeutic window for the treatment of infectious diseases.

Here, we focus on anti-infective chemotherapy (helminths, protozoa, bacteria, and viruses). We recapitulate the pros and cons of chemotherapeutic repurposing and then illustrate the two principal strategies, opportunistic vs. rational repurposing, with case stories of molecules that have been repurposed to, from, or within the field of anti-infective chemotherapy.

## 3. Pros of Drug Repurposing

Instead of starting from scratch with a new compound, drug repurposing leverages the knowledge, safety profiles, and established manufacturing processes of already approved or investigational drugs. This offers several advantages.

### 3.1. Saving Time and Money

The repurposing of drugs which have already undergone safety testing and regulatory approval in humans can save substantial time and resources compared to developing compounds de novo [23]. Starting from known bioactives, ideally with a known mode of action (with the caveat that the target might turn out to be a different one in the repurposed application), and known PK and ADME profiles, this may allow to bypass the initial stages of drug development, including preclinical toxicology studies and phase I clinical trials [24]. The average costs for developing a new drug are more than one billion dollars [25], and the time from discovery to application usually exceeds ten years [26]. Therefore, the repurposing of existing drugs is a highly attractive option to save money and shorten the time it takes from bench to clinic [27].

### 3.2. Expanding Therapeutic Opportunities to Address Unmet Medical Needs

Drug repurposing widens the range of therapeutic options available by identifying new indications for drugs that have already proven effective in different disease contexts. This approach capitalizes on the growing understanding of disease mechanisms and the potential for off-target effects of drugs, enabling the exploration of unanticipated therapeutic benefits [28]. The success stories of repurposed drugs, such as sildenafil (Viagra) repurposed for erectile dysfunction from its original use as an antihypertensive, and thalidomide repurposed for multiple myeloma from its initial indication as a sedative, highlight the transformative potential of drug repurposing [29]. Additionally, drug repurposing offers the possibility of finding effective treatments for diseases with limited or no approved therapies. This is particularly prominent in certain disease categories such as rare diseases, neglected tropical diseases (NTDs), and cancer [30].

## 4. Cons of Drug Repurposing

While the advantages of drug repurposing are obvious, there are also some drawbacks and pitfalls that need to be taken into consideration before venturing on a repurposing campaign.

### 4.1. Limited Specificity and Efficacy

The rational starting point towards a new drug is the definition of the target product profile (TPP), i.e., the properties a new therapy must have to make the desired impact. However, repurposed drugs may not be perfectly aligned with the TPP in the new context, be it with regard to pharmacodynamics, pharmacokinetics, toxicity, formulation, price, or stability. Thus, starting from a given molecule that is to be repurposed, rather than from a given TPP, involves the temptation to compromise on optimal use (the “Birmingham screwdriver”). Thus, chemotherapeutic repurposing not directed by a TPP might lead to a suboptimal outcome or limited therapeutic benefit [31].

### 4.2. Risk of Failure and Adverse Effects

The repurposed drug may not demonstrate the desired efficacy or safety profile in the new context, leading to treatment failures or adverse effects. The lack of preclinical and clinical data specific to the new indication increases the uncertainty associated with repurposing. This risk can even jeopardize the primary use when toxicity becomes apparent only during the alternative use [32]. One example was the clinical development of pafuramidine as a first oral drug to treat human African trypanosomiasis (HAT). In spite of promising results from a clinical phase II trial [33], the development of pafuramidine was stopped for toxicity issues that had only come up during a prolonged regimen tested in phase I with the aim of repurposing pafuramidine for Pneumocystis jirovecii infection.

### 4.3. Lack of Intellectual Property Protection

Repurposing existing drugs often involves the use of off-patent or generic drugs. This limits the potential for exclusivity and intellectual property protection, which can discourage pharmaceutical companies from investing in repurposing efforts, as they may not see significant financial return [29].

## 5. Challenges of Drug Repurposing for Infectious Diseases

Developing new drugs for infectious diseases involves particular challenges that impact the drug discovery and development process [34]. Some of the key challenges are described in the following paragraphs.

### 5.1. Complexity of Pathogens and Lack of Comprehensive Understanding of Pathogen–Host Interactions

The diverse mechanisms of infection and complex host–pathogen interactions make it challenging to identify suitable drug targets for anti-infective agents [35]. Many pathogens, including viruses, bacteria, protozoa, and nematodes, acquire high genetic diversity through mutation and recombination [36]. This genetic variability can lead to the emergence of drug-resistant mutants, making it difficult to develop drugs that effectively target all variants of the pathogen. Thus, continuous efforts are required to stay ahead of emerging resistance mechanisms [37]. Some pathogens, like *Neisseria* spp. or *Plasmodium* spp., undergo antigenic variation, where they change their surface proteins to evade the host’s immune response. This constant adaptation can hinder the identification of stable drug targets and vaccine candidates [38]. Other pathogens, like *Trypanosoma cruzi*, *Mycobacterium tuberculosis*, or herpes viruses, can establish latent or persistent infections, becoming dormant in the host’s cells and evading immune detection [39,40]. Curing such infections requires drugs that target the pathogen during both active and dormant phases. Some bacteria, like *Pseudomonas aeruginosa* and *Staphylococcus aureus*, can form biofilms that enhance drug resistance and impede drug penetration, making it challenging to eliminate infections [41].

### 5.2. Shortage of Effective Animal Models

Animal models that accurately mimic human infections are crucial for preclinical testing, but developing appropriate models for certain infectious diseases can be difficult. This can be due to involvement of complex host–pathogen interactions that are difficult to model accurately [42]. The lack of appropriate predictive models can impede drug development, as it becomes challenging to understand the disease mechanisms and assess the efficacy of potential drugs [43]. A breakthrough in antimalarial drug R&D was the establishment of a NOD-SCID mouse model for infection with Plasmodium falciparum that was suitable for drug efficacy testing [44].

### 5.3. High Risk of Clinical Failure

Clinical trials for anti-infectives often face high attrition rates due to insufficient efficacy, safety concerns, or challenges in patient recruitment [45]. Moreover, challenges in conducting clinical trials and follow-up in resource-limited settings and a lack of appropriate surrogate endpoints for infectious diseases contribute to the high failure rates [46].

### 5.4. Limited Market Incentives

Developing new therapies for infectious diseases is expensive, and the return on investment is lower compared to chronic diseases prevalent in developed countries, which discourages investment in infectious disease drug development [47]. This is partially due to the fact that infectious diseases predominantly affect low-income populations and may not provide sufficient financial incentives for pharmaceutical companies to invest in research and development [48].

## 6. Successful Examples of Drug Repurposing

In spite of all these challenges, a number of drugs have been successfully repurposed to provide new, effective treatments for infectious diseases [49]. Table 1 shows examples of molecules that were repurposed to the field of anti-infective chemotherapy from other indications. Such inbound repurposing to infectious diseases mainly happened from the field of cancer chemotherapy. This is explainable by the fact that parasites and tumor cells have a lot in common—first and foremost, a fast proliferation, which necessitates a high rate of nucleotide synthesis and cell division, and brings about a higher metabolic activity and the generation of radicals (Table 2). Thus, antifolates are used for the therapy of cancers, parasitoses, and bacterial infections. Antifolates are drugs that interfere with the synthesis of dTMP from dUMP, where folate acts as the methyl group donor [50,51]. Another common target in pyrimidine synthesis is dihydroorotate dehydrogenase (DHODH), which is being pursued in anticancer [52] and antimalarial [53] drug development. Also, proteasome inhibitors show chemotherapeutic potential against tumor cells [54] as well as parasites [55,56].

Many anticancer drugs are active against parasites, e.g., taxol and vinblastine against malaria parasites [57], or etoposide, methotrexate, and doxorubicin against trypanosomatids [58,59,60]. However, the TPPs of anticancer drugs are not the same as those of anti-infectives; they differ in particular regarding the final product’s cost-of-goods and safety. Risks and adverse reactions that may be acceptable during cancer therapy are not for infectious diseases such as malaria.

The molecules in Table 3 have been repurposed within the field of anti-infective chemotherapy, from one pathogen to another. This is favored if the two pathogens are phylogenetically close. A frequent scenario is the repurposing of veterinary drugs to human medicine to treat infections by parasitic nematodes. Indeed, all the drugs currently in human use have originally been developed as veterinary anthelminthics. In this context, it is relevant to note that human infections with gastrointestinal or filarial nematodes are neglected, whereas the related diseases in livestock and pet animals are not. A comparison of the closely related *Onchocerca volvulus* and *Dirofilaria immitis* should suffice. The former causes the neglected tropical disease river blindness, while the latter is the dog heartworm, for which the American Heartworm Society recommends the year-round prophylactic treatment of all dogs in the United States [61].

Drug repurposing is also possible between phylogenetically distant pathogens. Antifungals, for instance, can be repurposed against Trypanosoma and Leishmania because both, fungi as well as trypanosomatids, have ergosterol as the sterol component of their membranes and not cholesterol [62]. Several antibiotics have been repurposed to eukaryotic parasites that contain essential prokaryotic endosymbionts (e.g., Wolbachia in filarial nematodes) or remnants thereof (e.g., the apicoplast in malaria parasites).

## 7. Opportunistic Drug Repurposing for Infectious Diseases

With ‘opportunistic’ we refer to the serendipitous or accidental discovery of a new activity of an existing drug against a specific infectious pathogen that usually emerges from phenotypic screening campaigns [63]. It makes use of the advantages of repurposing without having a particular mode of action in mind or a prior knowledge of a validated molecular target, which is the case for most of the antiparasitic drugs that are available today [64]. There have been several examples that demonstrate the power of opportunistic repurposing for hit and early lead identification, up to providing new treatment options. For instance, the screening of the NIH library of approved molecules against various pathogens has identified clomiphene and toremifene (selective estrogen receptor modulators) as potent inhibitors of *Zaire ebolavirus* infection in vitro and in vivo [65]. The response appeared not to be associated with estrogen signaling, but rather with an off-target effect where the compounds interfere with viral entry and likely affect the triggering of fusion [65]. Another example is the inhibition of Zika virus proliferation, and of virus-induced cytopathic effects, in glial cell lines and human astrocytes by the macrolide antibiotic azithromycin [66]. The antiarthritic metallodrug auranofin exhibits potent antileishmanial activity [67] and macrofilaricidal activity through the inhibition of the redox enzymes thioredoxin reductase and thioredoxin glutathione reductase [68].

Some phenotypic screenings were customized for a particular TPP, such as the screening of compounds known to permeate the blood–brain barrier, aiming to identify potential treatments for late-stage HAT, when the trypanosomes have invaded the patient’s cerebrospinal fluid [69,70]. Moreover, opportunistic repurposing has also been considered for disease-targeted collections. For instance, the Malaria Box is a collection of compounds assembled by the Medicines for Malaria Venture (MMV) to support drug discovery against malaria [71]. While the primary focus obviously is malaria, some of the compounds have also been screened against other pathogens to explore their potential as broad-spectrum antiparasitic agents. This has returned potent hits against kinetoplastids [72], toxoplasmosis [73], amoebiasis [73], candidiasis [74], schistosomiasis [75], echinococcosis [76], and several other diseases [77].

## 8. Rational Drug Repurposing for Infectious Diseases

Rational repurposing utilizes the knowledge of the evolutionary relationship or the lifestyle of pathogens to identify vulnerable points of attack that are common to several pathogens but absent from their hosts. By understanding the shared biological mechanism or target among different pathogens, drugs that have been developed for one may be used for another. Based on the evolutionary origin of the shared point of attack, this can be further classified into divergent and convergent.

### 8.1. Divergent Target/Mode of Action

Divergent drug targets have a common ancestry, and they are present in several pathogens but absent in the human host. Typical examples are the enzymes of the non-mevalonate pathway (the MEP/DOXP pathway) for the synthesis of isopentenyl pyrophosphate (IPP) from pyruvate and glyceraldehyde-3-phosphate. This pathway is absent from animals, which use the mevalonate pathway to synthesize IPP from three molecules of acetate. IPP is the precursor of isoprenoids and thus essential for cell function. The MEP pathway is utilized by many bacteria, including pathogens such as *Salmonella* and *Mycobacterium*; it is also present in cyanobacteria and in the chloroplast of plants [78,79]; and it occurs in the apicoplast, the plastid-like non-photosynthetic organelle of apicomplexan parasites such as *Plasmodium* spp. and *Toxoplasma gondii* [80]. This apicoplast is the result of a secondary endosymbiosis, where a heterotrophic eukaryote has engulfed a red algal ancestor, retaining it as a plastid [81,82]. The production of IPP by the apicoplast is essential for the survival of *P. falciparum* inside red blood cells [83,84,85], which explains why antibiotics that target apicoplast functions possess antimalarial activity. A good example is fosmidomycin, a natural compound from *Streptomyces* spp. [86,87,88] that inhibits the rate-limiting enzyme of the MEP pathway, 1-deoxy-D-xylulose-5-phosphate reductoisomerase (DXR) [79,89,90,91]. Thanks to the fact that fosmidomycin had already been tested in humans as an antibiotic, it could be readily repurposed for antimalarial combination therapy [92].

The finding that both plants and malaria parasites have an evolutionary link to cyanobacteria may explain the antimalarial activity of herbicides: glyphosate, a well-characterized inhibitor of the shikimate pathway [93], the microtubule inhibitors dinitroaniline and phosphorothioamidate [94], and the inhibitors of serine hydroxymethyltransferase [95]. This concept was chemically explored by screening commercial herbicides against *P. falciparum* [96]. Repurposing the other way around, where antimalarial lead compounds could offer a starting point for the discovery of new herbicides, has also been suggested [97,98].

Another example for the divergent mode of action is the repurposing of antibiotics for filarial pathogens based on their action against *Wolbachia* [99]. *Wolbachia* is a type of endosymbiotic bacterium that resides in the cells of many filarial parasites, including those responsible for lymphatic filariasis and onchocerciasis [100]. The presence of *Wolbachia* is crucial for the survival and reproduction of these parasites by providing essential nutrients and metabolites; *Wolbachia* also modulates the host’s immune response and contributes to inflammatory pathologies associated with filarial infection [101]. Targeting *Wolbachia* offers a unique opportunity to disrupt the symbiotic relationship between the bacterium and the nematode, leading to parasite sterility, death, and clearance [102]. Several antibiotics have been identified as effective in targeting *Wolbachia* within filarial parasites [103]. The most extensively studied is doxycycline, which acts by inhibiting protein synthesis in *Wolbachia*, leading to its clearance from the host cells [104]. Other antibiotics with activity against *Wolbachia* include minocycline, rifampicin, azithromycin, and corallopyronin A [105,106]. Clinical trials and field studies have demonstrated the efficacy of antibiotics, particularly doxycycline. The treatment resulted in a decrease in microfilariae (the larval stages of the parasite) in the blood, a lower adult worm viability, and a reduced transmission of the parasite by the insect vectors. Moreover, the treatment has been associated with an improvement in lymphedema and other disease-related symptoms [107,108]. Thus, combining antibiotics with standard antifilarial drugs, such as ivermectin or diethylcarbamazine, has shown promising effects in reducing worm burden and microfilarial load in patients and also allows for a shorter treatment regimen [109].

Nitroimidazoles are broad-spectrum antimicrobials for micro- and anaerobic pathogens. They are prodrugs that cause damage through the formation of radicals in a cascade of redox reaction [110], which begins with the reduction of the nitro group. This requires a reductase plus an electron donor of sufficiently low redox potential. Unlike their aerobic host, micro- and anaerobic pathogens in the mammalian gut possess a specific enzyme, pyruvate:ferredoxin reductoisomerase (PFOR), which plays a key role in prodrug activation. PFOR reduces the nitro group of nitroimidazoles such as metronidazole or tinidazole [111]. The presence of oxygen in the environment competes with nitroimidazoles, leading to a decrease in the reductive activation of the drug and an increase in the catalytic recycling of the activated drug [112]. Originally, nitroimidazole drugs were primarily used to treat infections by *Bacteroides* spp., *Clostridium* spp., and anaerobic streptococci. Subsequently, they were repurposed against *Helicobacter pylori* [113]. In addition, they are effective against protozoan parasites, including *Giardia intestinalis*, *Trichomonas vaginalis*, and *Entamoeba histolytica* [114,115,116], which also possess PFOR genes. Metronidazole is the first-line treatment for trichomoniasis and amoebiasis; it is also being used for giardiasis, but threatened by the emergence of drug resistance [117]. In trypanosomatids, nitro-drugs such as benznidazole, nifurtimox, or fexinidazole are reductively activated by type-I nitroreductase [118,119,120], another redox enzyme that is absent from mammalian cells.

### 8.2. Convergent Target/Mode of Action

Convergent targets are shared among different species due to convergent evolution, where pathogens from separate lineages have independently evolved similar traits or strategies to adapt to similar micro-environments and challenges in their hosts [121]. Thus, convergent targets reflect a different evolutionary origin, but the same life-style. For instance, all parasites that consume hemoglobin have to deal with free heme and iron that are released, generating reactive oxygen species (ROS) and reactive nitrogen species (RNS) [122]. During their intraerythrocytic development, malaria parasites break down large amounts of hemoglobin. Also, the blood fluke *Schistosoma mansoni* resides in the human bloodstream and feeds on erythrocytes. During the feeding process, the parasite releases proteolytic enzymes to degrade host hemoglobin, leading to radical production [123]. The production of radicals in malaria or schistosomiasis contributes to the pathogenesis of the diseases, causing oxidative damage and triggering an immune response in the host, leading to inflammation and tissue damage [124]. However, parasites also possess an efficient antioxidant defense mechanism through the glutathione- and thioredoxin-dependent systems [125]. The induction of oxidative stress, either through the production of reactive oxygen species (ROS) or the disruption of redox homeostasis in the infected erythrocyte, is a key mechanism of action for several antimalarial drugs (e.g., chloroquine, artemisinin, and atovaquone) [126,127]. The repurposing of antimalarials to schistosomiasis is based on the rationale that both parasites degrade hemoglobin [128,129]. Clinical trials of artemether against *S. mansoni* showed promising results [130].

In the context of biofilm formation, where the protective environment hinders the effective treatment of both tumors and bacterial infections, convergent targets can offer a promising avenue for intervention (Table 2). One such compound that illustrates the concept of convergent targeting is toremifene. Toremifene, initially developed as an estrogen receptor modulator for breast cancer treatment [131], has shown potential in disrupting bacterial and fungal biofilms. In bacterial biofilms, toremifene can disrupt the structural integrity of the matrix and enhance the susceptibility of bacteria to conventional antibiotics [132]. This class of selective estrogen receptor modulators, the triphenylethylenes, has shown a wide range of activity against medically important human pathogens, including bacteria, fungi, parasites, and viruses [133].

## 9. Repurposing of Anti-Infectives to Other Fields

It is clear from the examples discussed above that most cases of repurposing happened from the economically larger towards the smaller market, e.g., from cancer chemotherapy to that of parasitic infections. However, the repurposing of antiparasitic drugs has also been pursued, in particular with the COVID-19 pandemic (which, however, has also shown some of the pitfalls of repurposing [134,135]; also see Cons above).

### 9.1. Antiparasitics against COVID-19

The best known case is the use of the antimalarials chloroquine and hydroxychloroquine in the treatment of COVID-19-associated pneumonia [136,137]. Studies have demonstrated that chloroquine exhibited significant efficacy in suppressing viral replication, displaying an effective concentration (EC_90_) value of 6.90 μM [138]. This was a case of rational repurposing based on chloroquine’s established mechanism of action that involves obstructing virus infection through the elevation of endosomal pH and the disruption of the glycosylation of the cellular receptor for SARS-CoV-2. It is also speculated that the immunomodulatory properties of the drug could augment the antiviral effect in vivo [138]. In contrast, the outcome of clinical trials was controversial. Many large randomized controlled trials showed no mortality benefit of chloroquine or hydroxychloroquine for hospitalized COVID-19 patients [139,140,141]. Finally, in June 2020, the United States’ Food and Drug Administration revoked the authorization for emergency use of chloroquine and hydroxychloroquine to treat COVID-19 patients [142].

Probably the oldest antiparasitic drug still in use today, suramin, demonstrated the inhibition of SARS-CoV-2 replication and possibly acts on early steps of the replication cycle, preventing the binding or entry of the virus [143]. Suramin was shown to be a potent inhibitor of the SARS-CoV-2 RNA-dependent RNA polymerase (RdRp), blocking the binding of RNA to the enzyme, and was found to be 20-fold more potent than remdesivir [144]. Suramin had been shown to inhibit RNA-dependent DNA polymerase (reverse transcriptase) already in 1979 [145]. Clinical studies to support an effectiveness of suramin against COVID-19 are lacking.

Nitazoxanide, an antiparasitic drug for diarrhea and enteritis triggered by *Cryptosporidium* spp. and *G. intestinalis* of known antiviral activity [146], inhibited SARS-CoV-2 replication in Vero E6 cells at a low micromolar concentration [138,147]. However, the resolution of symptoms in patients did not differ between nitazoxanide- and placebo-treated groups after 5 days of therapy [148]. The antiparasitic drug ivermectin is an additional example of the COVID-19 repurposing efforts [149]. The antiviral potential of ivermectin had been recognized in an opportunistic repurposing screen of randomly selected bioactives for the inhibition of importin αβ-mediated nuclear import [150]. Ivermectin was subsequently shown to inhibit the replication of HIV and Dengue virus [151], and it also inhibited SARS-CoV-2 replication in cell cultures [152]. However, clinical studies revealed that treatment with ivermectin did not result in a lower incidence of hospital admissions or of prolonged emergency department observation among outpatients with an early diagnosis of COVID-19 [153]. The repurposing attempts for COVID-19 underline the importance of making sure, before going to clinical trials, that not only the activity and tolerability of a drug candidate match the new TPP, but also its pharmacokinetics and pharmacodynamics.

### 9.2. Repurposing to Other Fields

The repurposing of antiparasitics to fields other than infectious diseases has also occurred; examples are shown in Table 4. Suramin, originally developed for Nagana and sleeping sickness, was tested for the treatment of cancers and autism [154], and it might also have potential for snakebite [155]. Azithromycin, a macrolide antibiotic, has shown anti-inflammatory properties and has improved lung function in patients with cystic fibrosis. Used as an adjunct therapy, it reduced pulmonary exacerbations [156]. Dapsone, an antibiotic primarily used for leprosy and certain infectious diseases, has been repurposed for inflammatory bowel disease (IBD), particularly in Crohn’s disease and ulcerative colitis, due to its anti-inflammatory properties [157]. However, the potential selection for resistant gut microbiota is of concern when repurposing antibiotics to treat chronic diseases.

In the field of vaccines, the *Mycobacterium bovis* Bacillus Calmette et Guérin (BCG) vaccine was originally developed in 1921 as a vaccine against active tuberculosis. The BCG vaccine later proved to possess activity against bladder cancer—due to its immunogenicity, and possibly also due to cytotoxic effects on tumor cells [158]. The exact mechanisms mediating the antitumor effect of the BCG vaccine remain to be elucidated [159,160]. Even so, the BCG vaccine has become an important adjuvant therapy for the treatment of high-risk non-muscle invasive bladder cancer, reducing the risk of cancer progression and recurrence [161].

The cosmetic industry has also benefited from drug repurposing. Botulinum toxin, or Botox, is a notable example that revolutionized the field of cosmetic enhancement. Botulinum toxin, a neurotoxic protein produced by the bacterium *Clostridium botulinum*, was initially investigated for its therapeutic effects in various medical conditions, such as muscle spasms, strabismus (crossed eyes), and cervical dystonia. By inhibiting the release of acetylcholine, botulinum toxin reduces the repetitive facial muscle contractions that lead to the formation of wrinkles, allowing the skin to appear smoother [162,163]. Eflornithine (difluoromethyl ornithine) is an inhibitor of ornithine decarboxylase, the first and rate-limiting enzyme in polyamine biosynthesis. Eflornithine had originally been developed as an anticancer agent and was subsequently repurposed to become the standard treatment of late-stage West African human trypanosomiasis (*gambiense* HAT). Based on the serendipitous discovery that eflornithine treatment leads to hair loss [164,165], it is now also being used in a crème to prevent the unwanted growth of facial hair, marketed as Vaniqa [166].

## 10. Future Directions in Drug Repurposing for Infectious Diseases: Artificial Intelligence (AI) and Machine Learning (ML)

Advanced computational methods, such as AI and ML, are currently exploited to accelerate drug discovery and repurposing [167]. These technologies enable the analysis of the large datasets emerging from genomics, proteomics, chemoinformatics, and clinical data, to identify potential repurposing candidates and predict drug–target interactions [168,169]. This data-driven approach utilizes computational algorithms that can consider drug–drug as well as disease–disease similarities, and the similarity between target proteins, chemical structures, and gene expression profiles [170,171]. In addition, AI and ML can help to prioritize drug candidates, optimize dosing, and identify novel drug combinations for infectious diseases. Regarding drug repurposing, AI-based tools have been utilized to accelerate drug discovery in the COVID-19 pandemic [172,173]. Recently, deep learning-guided discovery enabled the identification of abaucin as a new antibiotic molecule with highly selective activity against the Gram-negative bacterium *Acinetobacter baumannii* [174]. Overall, AI and ML are revolutionizing drug repurposing for infectious diseases by leveraging data analysis, prediction models, and optimization techniques. These technologies have the potential to identify effective treatments more rapidly, helping to address urgent medical needs during outbreaks and improving the output of drug discovery pipelines.

## Figures and Tables

**Table 1 molecules-29-00635-t001:** Inbound repurposing of drugs to the field of infectious diseases.

Molecule	Original Purpose	Envisaged New Purpose
Cisplatin	Cancer	Antibacterial
Eflornithine	Cancer	Human African trypanosomiasis
Gallium nitrate	Cancer	Pseudomonas aeruginosa
Miltefosine	Cancer	Leishmaniasis
Mitomycin C	Cancer	Microbial persisters
Niclosamide	Snail control	Antibacterial, antiviral, antifungal
Thalidomide	Morning sickness, soporific	Leprosy
Toremifene	Cancer	Bacterial biofilm

**Table 2 molecules-29-00635-t002:** Similarities between parasites and tumor cells, and how they can be exploited as drug targets.

Shared Property	Emerging Target/MoA	Drug
High rate of cell division	Microtubule spindle	Mebendazol
Fast rate of DNA synthesis	Topoisomerase	Quinolones
High demand of nucleotides	Pyrimidine synthesis	Antifolates, DHODH inhibitors
High rate of protein turnover	Proteasome	Proteasome inhibitors
High metabolic rate	Glycolysis	Antimycin A
Enhanced redox metabolism	Activation of prodrugs	Artemisinin
Cellular signaling pathways	Protein kinases	Sunitinib

**Table 3 molecules-29-00635-t003:** Drug repurposing within the field of infectious diseases.

Molecule	Original Purpose	Envisaged New Purpose
Albendazole	Veterinary helminthoses	Human helminthoses
Amphotericin B	Antifungal	Leishmaniasis
Artemether	Malaria	Schistosomiasis
Clindamycin	Antibacterial	Malaria
Closantel	Anthelmintic	MRSA
Doxycycline	Antibacterial	Filariasis, malaria
Fosmidomycin	Antibacterial	Malaria
Ivermectin	Veterinary helminthoses	Human helminthoses
Levamisole	Anthelmintic	Colon cancer
Nifurtimox	Chagas disease	HAT
Paromomycin	Antibacterial	Leishmaniasis
Pentamidine	African trypanosomiasis	Balamuthiasis, leishmaniasis
Posaconazole	Antifungal	Chagas disease
Sulphonamides	Antibacterial	Malaria
Suramin	African trypanosomiasis	Onchocerciasis, antiviral
Telacebec	Tuberculosis	Buruli ulcer, leprosy

**Table 4 molecules-29-00635-t004:** Outbound repurposing of drugs from the field of infectious diseases.

Molecule	Original Purpose	Envisaged New Purpose
Allopurinol	Leishmaniasis	Gout
Artemisinin	Malaria	Cancer
Eflornithine	African trypanosomiasis	Hirsutism
Ivermectin	Anthelminthic	Mosquito control
Minocycline	Antibacterial	Neurodegenerative disorders
Pentamidine	African trypanosomiasis	Cancer
Suramin	African trypanosomiasis	Cancer, snake bite, autism

## Data Availability

Not applicable.

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
