# Peer review of "Drug Repurposing in the Chemotherapy of Infectious Diseases"

_molecules, 2024, doi:10.3390/molecules29030635_

Round 1
Reviewer 1 Report
Comments and Suggestions for Authors
The authors present a comprehensive review about the drug repurposing in chemotherapy and infectious disease. The idea is good, and the structure is well-organized. However, there is a few concerns to be addressed.
Since the main topic is drug repurposing. It seems that the first section “Repurposing in Nature” is irrelevant to the topic. Instead, more reviews should be given to the background about the drug repurposing.
The authors give the pros and cons of drug repurposing in Section 3 and 4. Please provide more examples or applications in chemotherapy and infectious disease.
Typically, drug repurposing review will provide the methods and approaches in drug repurposing. The authors haven’t included this information. Could the authors give the short review section about the approaches used in chemotherapy and infectious disease?
Comments on the Quality of English LanguageNA
Author Response
We wish to thank the Reviewer for their constructive criticism. Please find below our response and attached a version of the manuscript where all the changes are highlighted in yellow (file molecules-2805752_R1_highlighted.pdf).
Since the main topic is drug repurposing. It seems that the first section “Repurposing in Nature” is irrelevant to the topic. Instead, more reviews should be given to the background about the drug repurposing.
We admit that the first section is somewhat out of the box. We think that it provides a refreshingly new angle on a topic that has been discussed by many others already. However, we have added four more reviews on drug repurposing as requested (line 65).
The authors give the pros and cons of drug repurposing in Section 3 and 4. Please provide more examples or applications in chemotherapy and infectious disease.
More examples are provided in Tables 1, 3 and 4 and the corresponding text.
Typically, drug repurposing review will provide the methods and approaches in drug repurposing. The authors haven’t included this information. Could the authors give the short review section about the approaches used in chemotherapy and infectious disease?
We explain the different approaches in terms of their rationale, i.e. opportunistic repurposing to save time vs. repurposing based on an evolutionary hypothesis (convergent or divergent). Explaining the drug research and development pipeline is beyond the scope of this review.
Reviewer 2 Report
Comments and Suggestions for Authors
The present review outlines the development and research performed in drug repurposing of chemotherapeutics in the treatment of virus, bacterial and parasitic infections.
The introduction into this field is well described by the definition of repurposing itself from a biological and biochemical point of view. Afterwards, drug repurposing is described with few examples given, before pros, cons, challenges and successfull examples of this approach are given and discussed. The realization of drug repurposing is then described by opportunistic and rational approaches and examples are given in case of antiparasitics. Last also briefly computational and big data methods are discussed.
The review is well structured and written in very good English. The main topic of drug repurposing of chemotherapeutics against infectious diseases is outlined in a clear manner.
Some minor remarks:
1) the abbr. TPP is introduced in l. 128 and l. 253 again
2) [40] in l. 178 non italic
3) Section "Repurposing of antiinfectives to other fields": The concepts of rational and opportunistic approaches are introduced in section 7 and 8 but not applied in section 9. If applicable (and known) it would be interesting to the reader to know if the introduced drugs e.g. in the development of COVID-19 drugs are classifiable into one of the introduced concepts.
Summarized the review is very interesting and informative to the interested reader and broadly summerizes the field field of drug repurposing. Therefore, the publication is recommended after minor revision or if point 3 is not applicable as it is.
Author Response
We wish to thank the Reviewer for their constructive criticism. Please find below our response and attached a version of the manuscript where all the changes are highlighted in yellow (file molecules-2805752_R1_highlighted.pdf).
1) the abbr. TPP is introduced in l. 128 and l. 253 again
Line 253 was corrected (only the abbreviation was kept).
2) [40] in l. 178 non italic
Corrected.
3) Section "Repurposing of antiinfectives to other fields": The concepts of rational and opportunistic approaches are introduced in section 7 and 8 but not applied in section 9. If applicable (and known) it would be interesting to the reader to know if the introduced drugs e.g. in the development of COVID-19 drugs are classifiable into one of the introduced concepts.
Information on the rational basis of repurposing was added to the section on repurposing of antiinfectives to other fields.
